

# Extended range forecasting of stream water temperature with deep learning models

Ryan S. Padrón[1,2], Massimiliano Zappa[1], Luzi Bernhard[1], Konrad Bogner[1]

[1]Research Unit Mountain Hydrology and Mass Movements, Swiss Federal Institute for Forest, Snow and Landscape
Research WSL, Birmensdorf, Switzerland
[2]Institute for Atmospheric and Climate Science, ETH Zurich, Zurich, Switzerland

*Correspondence to*: Ryan S. Padrón (ryan.padron@wsl.ch)

**Abstract.** Stream water temperatures influence water quality with effects on aquatic biodiversity, drinking water provision, electricity production, agriculture, and recreation. Therefore, stakeholders would benefit from an operational forecasting
service that would support timely action. Deep learning models are well-suited to provide probabilistic forecasts at individual stations of a monitoring network. Here we train and evaluate several state-of-the-art models using 10 years of data from 54 stations across Switzerland. Static catchment features, time of the year, meteorological observations from the past 64 days, and their ensemble forecasts for the following 32 days are included as predictors in the models to estimate daily maximum water temperature over the next 32 days. Results show that the Temporal Fusion Transformer (TFT) model
performs best with a Continuous Rank Probability Score (CRPS) of 0.70 ºC averaged over all lead times, stations, and 90 forecasts distributed over 1 year. The TFT is followed by the Recurrent Neural Network Encoder – Decoder with a CRPS of 0.74 ºC, and the Neural Hierarchical Interpolation for Time Series with a CRPS of 0.75 ºC. These deep learning models outperform other simpler models trained at each station: Random Forest (CRPS = 0.80 ºC), Multi-layer Perceptron neural network (CRPS = 0.81 ºC), and Autoregressive linear model (CRPS = 0.96 ºC). The average CRPS of the TFT degrades
from 0.38 ºC at lead time of 1 day to 0.90 ºC at lead time of 32 days, largely driven by the uncertainty of the meteorological ensemble forecasts. In addition, TFT water temperature predictions at new and ungauged stations outperform those from the other models. When analyzing the importance of model inputs, we find a dominant role of observed water temperature and future air temperature, while including precipitation and time of the year further improve predictive skill. Operational probabilistic forecasts of daily maximum water temperature are generated twice per week with our TFT model and are
publicly available at https://www.drought.ch/de/impakt-vorhersagen-malefix/wassertemperatur-prognosen/. Overall, this study provides insights on the extended range predictability of stream water temperature, and on the applicability of deep learning models in hydrology.

## 1 Introduction

The services provided by streams and rivers are conditioned by water quantity as well as quality (van Vliet et al., 2017,
2023). For water quality, temperature is a key and highly sensitive variable as recognized by scientists, practitioners, and





regulators (e.g. Arora et al., 2016; Hannah et al., 2008; Hannah and Garner, 2015; Johnson et al., 2024; Webb, 1996). Water temperature affects growth, reproduction, distribution, health, and survival of aquatic life (e.g. Alfonso et al., 2021; Booker et al., 2022; Elliott and Elliott, 2010; Hannah and Garner, 2015; Little et al., 2020; Singh et al., 2024), as well as dissolved oxygen (Chapra et al., 2021) and nutrient cycling (Comer-Warner et al., 2019; Johnson et al., 2024). Economic and societal

aspects of electricity production, drinking water provision, recreation, and tourism are also conditioned by water temperature (e.g. Michel et al., 2020; Ouellet et al., 2020; van Vliet et al., 2013). As we undergo the effects of the human-induced climate crisis, more frequent and new challenges related to changes in water temperature are expected (Caretta et al., 2022; Ficklin et al., 2023; Hardenbicker et al., 2017; Michel et al., 2022; van Vliet et al., 2023).

Data-informed decisions are essential to adequately address present and future challenges associated to stream water temperature. The ongoing expansion of monitoring networks and water temperature research (Ouellet et al., 2020) are steps in the right direction. Whereas this largely applies to Europe and North America, throughout most of the world water temperature data remain sparse, restricted, and fragmented in space and time (Ficklin et al., 2023; Hannah et al., 2011; van Vliet et al., 2023). In addition, observed and projected long-term stream water temperature regional warming trends (Arora

et al., 2016; Hardenbicker et al., 2017; Kelleher et al., 2021; Michel et al., 2020, 2022) are often insufficient for stakeholders that require timely information from individual streams at high temporal resolution. The monitoring network of the Swiss Federal Office for the Environment aims to satisfy these requirements and is used in this study to predict daily maximum water temperature across a wide range of river stations.

Stream temperatures across regions are primarily determined by climate, with air temperatures having a dominant effect. Nevertheless, the relationship between air and water temperature at individual catchments is mediated by the local meteorology, hydrology, and watershed characteristics (Hannah and Garner, 2015; Wade et al., 2023). Typically, air temperature and runoff are used to predict stream temperature (e.g. Qiu et al., 2021; Toffolon and Piccolroaz, 2015; Zhu and Piotrowski, 2020), and less often solar radiation, precipitation, and base flow (groundwater contribution) have also been

considered (Arora et al., 2016; Feigl et al., 2021; Wade et al., 2023). The response of water temperature to changes in these environmental conditions can vary according to catchment characteristics (Wade et al., 2023). In Switzerland, contributions of glacier/snow meltwater are relevant as they reduce the sensitivity of stream temperature to an increase in air temperature (Michel et al., 2020). Another important aspect is the shorter hydrologic residence time of small steep catchments, which hinders their ability to accumulate heat compared to larger flatter catchments that often encompass lakes. Lastly, direct

human impacts from reservoir management, water withdrawal, wastewater discharge, urbanization, etc, further increase the complexity of processes influencing stream temperatures (Ficklin et al., 2023).

Statistical data-driven models are used to predict stream water temperature in an operational setting relevant for decision-making, given that it is usually not possible to meet the high data requirements of process-based models to solve the energy



transfer equations to and from the river (Benyahya et al., 2007; Dugdale et al., 2017; Feigl et al., 2021; Zhu and Piotrowski, 2020). Statistical models estimate water temperature as a function of related covariates and range from simple linear (auto)regression models to novel deep learning model architectures (Tripathy and Mishra, 2023). The results from Feigl et al. (2021) for 10 river stations in Austria show an average mean absolute error of 0.44 °C for the best performing machine learning models out of set including: Random Forest, XGBoost, Feed-Forward neural networks, and Long Short-term

Memory (LSTM) recurrent neural networks. These machine learning models clearly improve the prediction of daily water temperature compared to the average mean absolute error of 1.24 °C for linear regression and 0.76 °C for the Air2stream hybrid model, which combines a physically based structure with a stochastic calibration of the parameters (Toffolon and Piccolroaz, 2015). In another study with data from 8 river stations in the United States, Switzerland and China, Qiu et al. (2021) found an average mean absolute error of 0.57 °C for a deep learning LSTM, which outperformed Air2stream, a

Random Forest model, and a Back Propagation neural network. To date there are novel deep-learning architectures with promising results for time series forecasting (Challu et al., 2023; Lim et al., 2021; Wen et al., 2018), which are yet to be applied to forecasting water temperatures (Tripathy and Mishra, 2023).

Here we evaluate the skill of three of these state-of-the-art deep-learning models for predicting daily maximum water

temperature in 54 river stations in Switzerland over the next 32 days since the start of the forecast against three more common and simpler models. An important innovation of these models is direct multi-horizon forecasting (i.e. the simultaneous prediction of multiple future time steps) instead of iteratively forecasting one day after another, which increases efficiency and robustness (Challu et al., 2023; Fan et al., 2019; Lim et al., 2021). Furthermore, some of these models produce probabilistic forecasts that are useful for risk management under uncertainty. We also assess the uncertainty

stemming from the forecasts of meteorological variables that are used as predictors of stream temperature. Counting with probabilistic stream temperature forecasts over the upcoming month allows users to optimize the timing of their actions, such as adaptation measures when facing extreme conditions (e.g. Ouellet-Proulx et al., 2017). The analysis of extend range forecasts is a novel aspect of our study that goes beyond the traditional focus on one-step-ahead forecasts. In this study we also analyse the extrapolation capabilities of the deep learning models to predict stream water temperature at new and

ungauged stations, as well as the predictive importance of model inputs and previous time steps. Lastly, we present an example of our operational forecasts with the best performing model.

## 2 Methods

### 2.1 Data

We use a subset of 54 stations from the Swiss Federal Office for the Environment monitoring network

(https://www.hydrodaten.admin.ch/en/) (Fig. 1 and Table S1). Our variable of interest is daily maximum stream water temperature (WT), and for the analysis we use data from 12/05/2012 until 31/12/2022. In addition, we employ



meteorological data, plus information on catchment characteristics and time of the year as relevant features for forecasting stream water temperature.

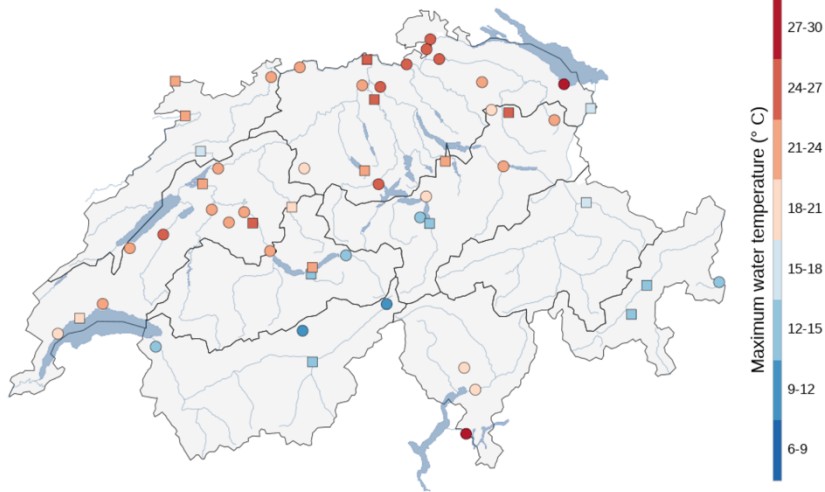

**Figure 1: Map of 54 stream water temperature stations in Switzerland used for the analysis.** The color bar indicates the observed mean annual maximum daily water temperature from 2012 to 2022. Squared markers indicate the subset of stations not used for training the models when evaluating the predictive skill at new and ungauged stations.

The area of the catchments ranges from 3.19 km² for station 2414 (Rietholzbach - Mosnang, Rietholz) to 34524 km² for station 2091(Rhein - Rheinfelden, Messstation), and the mean elevation from 503 m for station 2415 (Glatt - Rheinsfelden) to 2704 m for station 2256 (Rosegbach - Pontresina). Glacierized area fraction can reach up to 24.7% for station 2269 (Lonza - Blatten). Table S1 details the catchment characteristics from the 54 stations. To inform about the time of the year we define a date index (DI) as a sine function of the week of the year (woy) (Eq. 1), such that values range from near 0 at the end of January (winter) up to 1 at the end of July (summer). A modified date index as a predictor has been shown to improve the skill and/or training time of models (Feigl et al., 2021; Zhu and Piotrowski, 2020).

$$DI = \left| \sin\left(\pi * \left(woy - \frac{52}{12}\right)/52\right)\right| \qquad (1)$$

The considered meteorological variables include daily average near surface air temperature (AT), precipitation (P), and daily fraction of sunshine duration (SD). Gridded data of these variables is provided by the Swiss Federal Office for Meteorology and Climatology at a spatial resolution of 2 km. For simplicity we use for each station the time series of these variables from the single grid cell where the station is located, given the spatial coherence between neighboring grid cells. In addition to past observed values of these meteorological variables, their forecasts for the next 32 days are taken into account to forecast stream water temperature up to the same lead time. The meteorological forecasts correspond to a downscaled version of the Extended-range forecasts from the European Centre for Medium-Range Weather Forecasts (ECMWF), following the setup presented and used by Bogner et al. (2022) and Chang et al. (2023) for streamflow forecasting. We use the ensemble



forecasts with 51 members (1 control and 50 perturbed initial conditions) provided by ECMWF to estimate the increasing uncertainty with lead time.

## 2.2 Forecasting models

We use three deep-learning models: Recurrent Neural Network Encoder – Decoder (RNNED), Temporal Fusion
Transformer (TFT), and Neural Hierarchical Interpolation for Time Series forecasting (NHITS). In addition, we use three more common and simpler models: Autoregressive linear model with exogenous variables (ARX), Random Forest (RF), and Multi-layer Perceptron neural network (MLP) with one hidden layer to put our results into context with previous studies. Data from 2012 to 2021 is used for training the models, whereas data from 2022 is used for evaluating their predictions. For all deep-learning algorithms a single general model is trained with data from all 54 stations, and daily values for the next 32
days are predicted in a single forward pass. On the other hand, the ARX, RF, and MLP models are trained separately for each station and forecasts are produced iteratively one day at a time up to the 32 days lead time. TFT, NHITS, RF and MLP generate quantile forecasts of q2, q10, q25, q50, q75, q90, and q98; whereas RNNED and ARX predict a single best estimate. Source code and implementations of the deep-learning models we employ are publicly available in the PyTorch Forecasting documentation (Beitner, 2020). For RF we use the quantile regression forest implementation (Meinshausen,
2006) within the R package "ranger" (Wright and Ziegler, 2017), for MLP we use the R package "qrnn" (Cannon, 2024), and for ARX the linear model option in the scikit-learn Python tool.

RNNED is a sequence-to-sequence framework that uses LSTM to encode the history of the input sequence into a context vector and to recursively decode it into predictions (Cho et al., 2014; Hochreiter and Schmidhuber, 1997). Direct multi-
horizon forecasting is performed by the decoder generating a sequence of future predictions at once. The encoder is a RNN whose hidden state at the time step when the forecast starts is a summary $c$ of the whole input sequence, while the decoder is a RNN that generates predictions $y_t$ based on its hidden state $h_t$, the prediction of the previous time step $y_{t-1}$, and $c$. Note also that $h_t$ itself is a function of $y_{t-1}$ and $c$, as well as $h_{t-1}$.

TFT has an attention-based architecture that integrates information from any time step and captures long-term dependencies in the data through interpretable self-attention layers (Lim et al., 2021). The multi-head attention block adds interpretability by identifying relevant time steps within the encoder period. Local temporal processing is done with LSTMs in the encoder and decoder in a similar manner to the RNNED. The TFT also includes Variable Selection Networks that at each step inform about the importance of the individual predictors and allow the model to neglect irrelevant inputs. Gated Residual Networks
(GRN) consisting of two dense layers and an Exponential Linear Unit (ELU) activation function (Clevert et al., 2015) are used within the Variable Selection Networks and elsewhere in the model architecture as gating mechanisms to adapt the model's depth and complexity according to the task at hand. Note that attention-based approaches are computationally



demanding given that they can explicitly model the interaction between every pair of input-output elements (Challu et al., 2023).


Finally, NHITS follows a different approach characterized by multi-rate sampling of the input and hierarchical interpolation of the output to increase computational efficiency, particularly for long-horizon forecasts (Challu et al., 2023). The model is composed of stacks of MLP blocks, with each block capturing a different part of the time series, and with each stack dealing with a different frequency (timescale) of the time series. The stacks range from those with a smooth input and low

cardinality output to those with high frequency input and high cardinality output. The forecast over all lead times is assembled by summing the temporally-interpolated outputs of all blocks from all stacks.

For all three deep learning models we define the length of the encoder to be 64 days, whereas the forecasting horizon goes up to 32 days. We use 24 forecast creation times every 15 days during the year 2021 as our validation set when training the

models. During each epoch of the training, 30 batches of 64 encoder-decoder chains are randomly sampled from the set of time series composed by the data of all 54 stations between 2012 and 2020. At the end of each epoch, the model parameters are updated only if they reduce the average quantile loss over all selected quantiles, lead times, and forecast creation times of the validation set. The quantile loss (QL) is given by Eq. (2), where $q$ is a quantile, $y$ is the observed value, $\hat{y}$ is the prediction, and $(.)_+ = \max(0, .)$. Note that the QL averaged over all quantile levels is an approximation of the well-known

Continuos Ranked Probability Score (CRPS) (Fakoor et al., 2023; Laio and Tamea, 2007). In the case of RNNED, we use mean absolute error (MAE) instead of QL, because QL is not supported in the source code. Here we decide to stop the training of the models if the parameters are not updated during 60 consecutive epochs, suggesting that they have converged to an optimal value. In addition, to limit computing time we stop the training if a maximum number of 200 epochs is reached. Finally, given that the parameter optimization is not deterministic, we train each deep learning model 10 times with

a different random seed.

$$QL = q(y - \hat{y})_+ + (1 - q)(\hat{y} - y)_+ \qquad (2)$$

To train the ARX, RF and MLP models we use all data from 2012 to 2021, and their default settings. A least squares fit is

done for ARX, whereas QL is used when training the RF and MLP.

**2.3 Model features and hyperparameters**

The deep learning models include static catchment characteristics as predictors, given that a single model is fitted for all 54 stations. Here we use all the static features provided in Table S1, i.e. catchment area, mean elevation, glacierized fraction, station coordinates, and long-term average and standard deviation of the target variable (target center and scale). The primary

set of known time varying model features used for the analysis includes meteorological variables and time of the year, i.e.





AT, P, SD, and DI, as these are commonly available. Additional sets of model features that exclude predictors are also evaluated in section 3.3. Lastly, the models also include observed WT from the encoder period.

One important aspect that influences the predictions of machine learning models is the selection of hyperparameters (e.g.
(Feigl et al., 2021; Kraft et al., 2024). Therefore, we conduct hyperparameter tuning for each of the deep learning models using the Optuna framework (Akiba et al., 2019). Proposed hyperparameter values are iteratively sampled 25 times with a Tree-structured Parzen Estimator (Bergstra et al., 2011) that efficiently explores the hyperparameter space by focusing on the regions with the largest potential to improve the model skill, i.e. to reduce the validation loss. Additionally, hyperband pruning is used to stop early on the iterations with hyperparameters that won't improve skill (Li et al., 2018). Details on the
hyperparameters of each model, the explored hyperparameter space, and their tuned values are provided in Table S2.

The ARX, RF and MLP models are trained per station, and hence do not include static features. These models do not encode information from the recent past, so here we include lagged (observed or predicted) water temperature at time $t$-$1$ (lagWT) to predict water temperature at time $t$. The final set of features includes lagWT, AT, P, SD and DI. We set the RF to 500 trees
of unlimited depth, and the MLP to have 1 hidden layer with 2 hidden nodes.

## 2.4 Forecasting at new and ungauged stations

The capability of a model to generate skillful water temperature forecasts at new locations can be of great added value for stakeholders because long-term measurements are not available everywhere (Ouarda et al., 2022). Given that deep learning models are trained on a set of stations, instead of individual time series, we expect some transferability in space to locations
with similar conditions. To analyze the performance of the deep learning models at new stations with water temperature observations and at ungauged locations we consider models with three different setups: (A) our control model trained on all 54 stations, (B) a model trained on a subset of 34 stations with the same features as A, and (C) a model trained on a subset of 34 stations excluding past observations of water temperature from the features (i.e. ungauged). Finally, we compare the predictions of A and B, as well as A and C, across the subset of remaining 20 stations not used when training B and C.

We divide our data into two subsets of 34 and 20 stations with similar distributions of catchment area, mean elevation and glacierized fraction as indicated in Table S1. In addition, we note that for setup B we use an encoder normalizer instead of group normalizer when training the model. This means that the features include the average and standard deviation of water temperature (target center and scale) over the encoder period, as opposed to their long-term average as in A. Meanwhile for
setup C, given that we do not include past water temperature as a feature, we also do not include its average and standard deviation.





## 2.5 Model performance evaluation

We use observed stream temperature from the year 2022 (excluded from model training) and 90 forecasts distributed over the year to assess the predictive skill of the models (Table S3). We do so with the Continuous Ranked Probability Score
(CRPS), which is designed for ensemble forecasts (e.g. Jollife and Stephenson, 2012). The CRPS compares the cumulative distribution function (CDF) of the forecasts against the observations, and it corresponds to the mean absolute error for the case when a single value is predicted. Therefore, the units of the CRPS are those of the observed variable, with values closer to 0 corresponding to a better agreement between predictions and observations.

In our case, for each of the 51 driving meteorological forecasts, the models generate a probabilistic forecast of daily maximum water temperature given by quantiles q2, q10, q25, q50, q75, q90 and q98. First, for each set of quantiles we fit a normal distribution with mean $\mu = q50$ and standard deviation $\sigma = (q90 - q10) / (sn(q90) - sn(q10))$; where $sn(qX)$ corresponds to the quantile $X$ of the standard normal distribution (i.e. with a standard deviation of 1). We then sample 100 values from each of the 51 fitted normal distributions and fit a final normal distribution to these data with parameters $\mu_f$ and
$\sigma_f$. This final distribution is used when computing the CRPS according to Eq. 3, where $y$ is the observed value (i.e. daily maximum water temperature), $\Phi$ is the CDF of the standard normal distribution, and $\phi$ is probability density function of the standard normal distribution.

$$CRPS = \sigma_f \left( \frac{y - \mu_f}{\sigma_f} \left( 2\Phi\left(\frac{y-\mu_f}{\sigma_f}\right) - 1 \right) + 2\phi\left(\frac{y-\mu_f}{\sigma_f}\right) - \frac{1}{\sqrt{\pi}} \right) \qquad (3)$$

## 3 Results and discussion

### 3.1 Model performance comparison

We first compare the predictive skill of the models omitting the uncertainty stemming from the meteorological forecasts of AT, P, and SD, i.e. when using their observed values instead of their forecasts over the 32 days of the prediction horizon (Fig. 2). The TFT performs best with an average CRPS of 0.39 ºC over all random seeds, lead times, stations, and forecast
start dates. The other deep learning models RNNED (0.57 ºC) and NHITS (0.60 ºC) are next in line, closely followed by the RF (0.64 ºC) and MLP (0.66 ºC). For the simplest linear ARX model the CRPS degrades to 0.97 ºC, which is 0.58 ºC worse than that of the TFT. The deep learning models show little spread in their average CRPS across the 10 random seeds: 0.014 ºC for the TFT, 0.040 ºC for the RNNED, and 0.022 ºC for the NHITS. This suggests the models converged to optimal parameters during training. Regarding the run time needed on our server to train each model for 200 epochs, the longest is 44
minutes for the TFT with 286.1k parameters, followed by 21 minutes for the RNNED with 60.9k parameters, and 17 minutes





for the NHITS with 103.8k parameters. For the simpler models less than a minute is needed to train them at each station, and this can easily be done in parallel.

For all models the predictive skill decreases with lead time, as the influence of observed daily maximum water temperature on the predictions also reduces (Fig. 2a). The decrease in skill is particularly fast during the first 4 days for ARX, MLP and RF, suggesting that these models have a higher reliance on past water temperature for predicting future water temperature. Across stations, there is substantial variability in the CRPS for all models (Fig. 2b). As an example for the TFT, it ranges from 0.23 °C for station 2276 (Grosstalbach – Isenthal) to 0.74 °C for station 2068 (Ticino – Riazzino). No clear relationships emerge when evaluating predictive skill as a function of catchment characteristics such as area, elevation and glacierized fraction (Fig. S1). Lastly, we note that the predictive skill of the models also varies seasonally (Fig. 2c). The CRPS is lower in the winter when water temperature is colder and has lower day-to-day variability, whereas it is higher in the summer when the conditions are opposite.

It is likely that stream water temperature at several of the Swiss stations analyzed in this study is influenced by reservoir and lake management (Michel et al., 2020), and potentially also by water withdrawal and discharge from industry for example. Larger deviations between observed and forecasted values are expected when management decisions influencing water temperature take place, given that their timing can be largely arbitrary, and thus not captured by our models. This seems to be the case for stations 2068 (Ticino – Riazzino), 2084 (Muota – Ingenbohl), and 2351 (Vispa – Visp) with the highest CRPS as seen in Fig. 2b. These stations are highly influenced by hydropeaking from the release of large volumes of cold water from reservoirs at high elevation (Michel et al., 2020). It is noteworthy that the high CRPS values for forecast-start-dates between mid-April to mid-May (Fig. 2c) are mainly driven by high CRPS values at these affected stations (Fig. S2). Furthermore, during this time of the year 2022 the models underestimate the observed water temperatures (Fig. S3), which is a reasonable consequence if less cold water was released during the drought year 2022 compared to past years used to train the models.



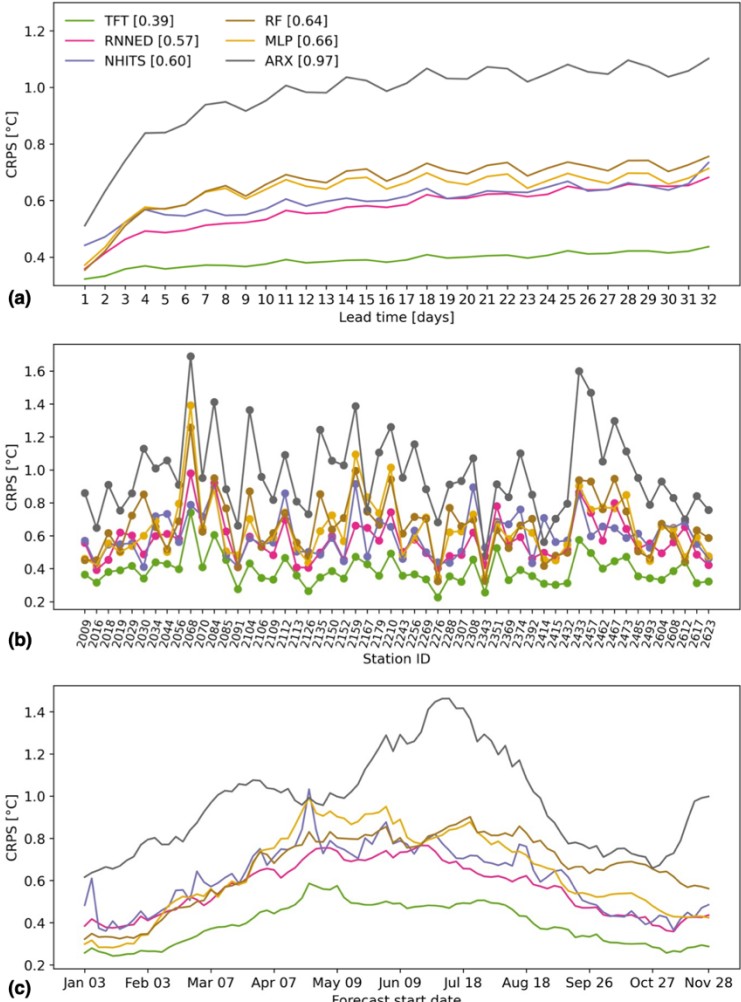

**Figure 2: Model comparison of predictive skill when omitting the uncertainty of meteorological forecasts.** The cumulative rank probability score (CRPS) of each model is shown as a function of **(a)** lead time averaged over all stations and forecasts, **(b)** station averaged over all lead times and forecasts, and **(c)** forecast start date averaged over all lead times and stations. The legend indicates the different models and their average CRPS over all 32 lead times, 54 stations, and 90 forecasts distributed over the year 2022.

Figure 3 compares the operational predictive skill of the models when using AT, P, and SD forecasts over the 32 days of the prediction horizon. The TFT remains the best performing model with an average CRPS of 0.70 ºC over all random seeds, lead times, stations, and forecast start dates, followed by RNNED with 0.74 ºC, NHITS with 0.75 ºC, RF with 0.80 ºC, MLP with 0.81 ºC, and ARX with 0.96 ºC. When comparing these results to those from Fig. 2, we note that the uncertainty of the meteorological forecasts contributes 0.31 ºC to the total disagreement between observations and TFT predictions, and less for the other models. Consequently, the uncertainty of the meteorological forecasts decreases the gain in predictive skill to be achieved when using the TFT compared to the other models. This is particularly the case for lead times beyond 5 days when there is often a strong decrease in meteorological predictability (Bauer et al., 2015).





There is an evident decrease in predictive skill of water temperature as lead time increases (Fig. 3a), which is directly related

to the ensemble meteorological forecasts being less accurate and having a larger spread for predictions further into the future. For all models, the CRPS at a lead time of 32 days is more than double its value at a lead time of 1 day. In the case of the TFT it increases from 0.38 ºC at lead time 1 to 0.90 ºC at lead time 32. In addition, we find that the uncertainty of the meteorological forecasts increases the CRPS variability across stations and forecast start dates, and can lead to other models outperforming the TFT in some cases (Figs. 3b and 3c).

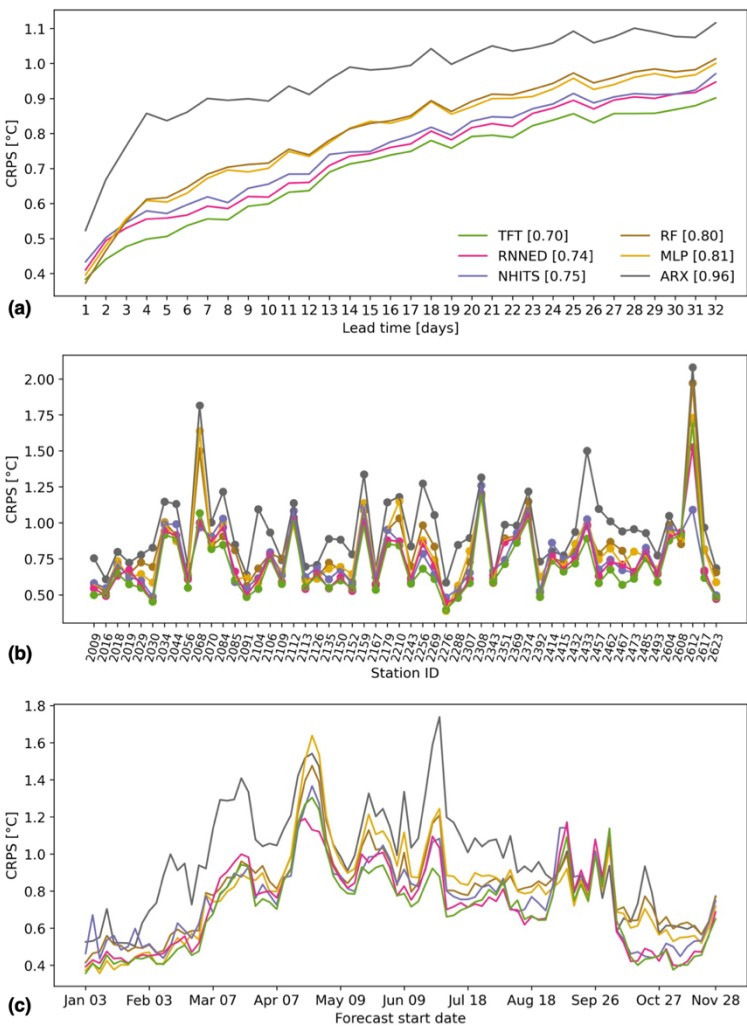


**Figure 3: Model comparison of predictive skill when including the uncertainty of meteorological forecasts.** The cumulative rank probability score (CRPS) of each model is shown as a function of **(a)** lead time averaged over all stations and forecasts, **(b)** station averaged over all lead times and forecasts, and **(c)** forecast start date averaged over all lead times and stations. The legend indicates the different models and their average CRPS over all 32 lead times, 54 stations, and 90 forecasts distributed over the year 2022.





High average CRPS values greater than 1 ºC – even for the TFT – occur at several stations: 2612 (Riale di Pincascia –
Lavertezzo), 2308 (Goldach - Goldach, Bleiche, nur Hauptstation), 2068 (Ticino – Riazzino), 2374 (Necker – Mogelsberg,
Aachsäge), and 2112 (Sitter – Appenzell), representing a large increase compared to their values in Figure 2 (except for
station 2068 influenced by reservoir management). Therefore, these high errors in predicted water temperature arise from
errors in the meteorological forecasts. Furthermore, the catchment area of the above-mentioned stations is less than 90 km$^2$

(except station 2068), and the catchment of station 2612 is particularly steep. These conditions are likely to make water
temperature more sensitive to changes in AT, P, and SD – i.e. the change in water temperature per unit change in the
meteorological predictors is greater (e.g. Wade et al., 2023). Figure S4 indeed shows a tendency for higher CRPS at stations
with smaller catchment area. Finally, the high CRPS values during April and September in Figure 3c result from the
contribution of most stations (with a strong influence from station 2612) (Fig. S5). This decrease in predictive skill was

caused by specific weather events that deviated strongly from the meteorological forecasts, particularly for lead times
beyond two weeks (Fig. S6).

## 3.2 Predictive skill at new and ungauged stations

Figure 4 compares the CRPS across 20 stations from predictions of deep learning models trained either including (setup A)
or excluding (setup B) data from this subset of stations. When forecasting water temperature at new stations on which the

models are not trained, results show that the TFT performs best with an average CRPS of 0.83 ºC across all 32 lead times, 20
stations, and 90 forecast start dates, followed by the NHITS with 0.92 ºC, and the RNNED with 1.11 ºC. The reduction in
prediction skill when extrapolating the models to new stations is 0.11 ºC for the TFT, 0.12 ºC for the NHITS, and 0.34 ºC for
the RNNED. The larger drop in skill of the RNNED occurs at more than half of the 20 stations and throughout most forecast
dates, with particularly large values at the highest elevation stations: 2256 (Rosegbach – Pontresina) and 2462 (Inn – S-

chanf). On the other hand, we note the good extrapolation capability of the TFT to new stations during the summer, with its
CRPS following closely that of the TFT trained on all 54 stations.



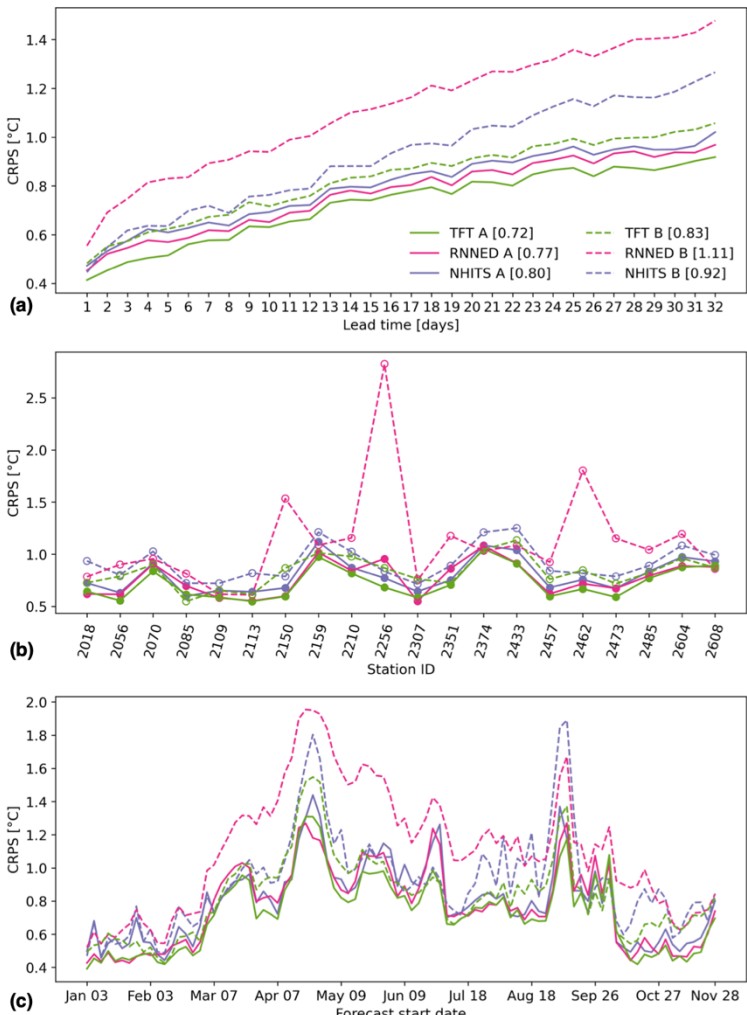

**Figure 4: Model comparison of predictive skill at new stations.** Continuous lines correspond to setup A when models are trained on data from all 54 stations, whereas dashed lines correspond to setup B when models are trained on data excluding the subset of 20 stations. The cumulative rank probability score (CRPS) of each model is shown as a function of **(a)** lead time averaged over subset of 20 stations and all forecasts, **(b)** station averaged over all lead times and forecasts, and **(c)** forecast start date averaged over all lead times and subset of 20 stations. The legend indicates the different models and their average CRPS over all 32 lead times, 20 stations, and 90 forecasts distributed over the year 2022.

There is a clear decrease in prediction skill across the 20 stations when forecasting with models that exclude information on past water temperature, and that are trained omitting the data from these stations (setup C) – as to represent ungauged stations (Fig. 5). Nonetheless, the TFT is still able to achieve an average CRPS of 1.04 ºC across all 32 lead times, 20 stations, and 90 forecast start dates, which corresponds to a reduction in prediction skill of 0.32 ºC compared to the model that includes past observations of water temperature and is trained using data from all 54 stations. Up to two weeks of lead time, the TFT average CRPS at ungauged stations is below 1 ºC. Furthermore, the CRPS of the TFT is lower than that of the RNNED and NHITS across almost all 20 stations and 90 forecast start dates. On the other hand, it is evident that the NHITS



model is not well-suited to generate predictions at ungauged stations as it strongly relies on past observations of the target variable to forecast future values.

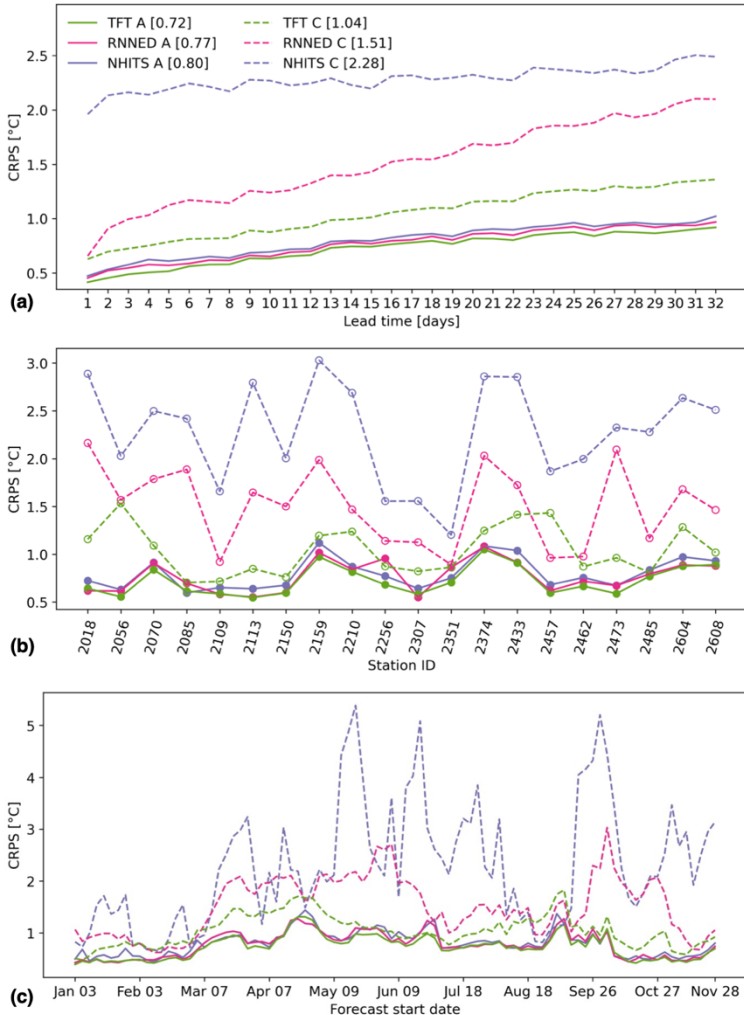

**Figure 5: Model comparison of predictive skill at ungauged stations.** Continuous lines correspond to setup A when models include past observations of water temperature as a feature and are trained on data from all 54 stations, whereas dashed lines correspond to setup C when models exclude past observations of water temperature as a feature and are trained on data excluding the subset of 20 stations. The cumulative rank probability score (CRPS) of each model is shown as a function of **(a)** lead time averaged over subset of 20 stations and all forecasts, **(b)** station averaged over all lead times and forecasts, and **(c)** forecast start date averaged over all lead times and subset of 20 stations. The legend indicates the different models and their average CRPS over all 32 lead times, 20 stations, and 90 forecasts distributed over the year 2022.

## 3.3 Predictive importance of model inputs

Here we focus on the TFT given that it outperforms the other models, and because of its built-in interpretability (Lim et al., 2021). For every forecast, the TFT directly outputs fractional importance weights (*importance*) that sum up to 1 for static features, for encoder features, and for decoder features separately. On average across the 10 random seeds and 54 stations,



results show similar *importance* values for all static features (Fig. S7). Target center (i.e. the long-term average of daily maximum water temperature) with 0.16 has the highest *importance*, whereas catchment area with 0.09 has the lowest *importance*. For individual stations, features with higher *importance* are typically those that differentiate them from other stations. For example, the importance of catchment glacierized fraction increases for stations with higher glacierized fraction.

Figure 6 shows WT has the highest *importance* among the encoder features with an average value of 0.35 over all random seeds, encoder time steps, stations, and forecasts. AT, SD, and DI have similar *importance* values of 0.17, 0.18, and 0.19, respectively, whereas P with 0.11 has the lowest. There is very low *importance* variability of the encoder features across time steps, stations and forecast start date. Nevertheless, we note a slightly higher *importance* of WT at larger catchments, and a small increase in the *importance* of AT for forecasts in July and August. For the decoder features, AT and DI have the highest average *importance* with a value of 0.33, followed by P with 0.20, and SD with 0.14. These weights are almost constant for all decoder time steps. Across stations, we find that approximately in half of them AT is slightly more important than DI, whereas the opposite is true for the remaining half. Also, there is a small but noteworthy increase in P *importance* at high elevation stations at the expense of AT. Lastly, results suggest that DI is generally more important for forecasts during the cold months, whereas the *importance* of AT, P, and SD increases during the warm months.

When interpreting the *importance* weights, it is useful to acknowledge that the model features are clearly not independent from one another. The meteorological features correlate with DI due to their seasonality. Higher SD is expected to coincide with higher AT, particularly during the warmer months. In addition, cloudiness influences the number of sunny hours and is a prerequisite for precipitation, thus relating SD and P. Consequently, the feature *importance* values can vary substantially every time the TFT model is trained with a different random seed (Fig. S8), even though the predictive skill hardly changes. In an operational context, the *importance* weights are nonetheless relevant to shed light on which meteorological conditions influence the model more every time a new forecast of daily maximum water temperature is generated.







**Figure 6: TFT feature importance.** The *importance* of each feature is shown as a function of **(a,d)** time step averaged over all stations and forecasts, **(b,e)** station averaged over all time steps and forecasts, and **(c,f)** forecast start date averaged over all time steps and stations. Encoder feature *importance* is shown in **(a,b,c)** and decoder feature *importance* in **(d,e,f)**. Note that in **(b)** stations are sorted by catchment area in ascending order, and in **(e)** stations are sorted by station elevation in ascending order. In all cases the *importance* is averaged across 10 models trained with different random seeds.

The TFT also provides fractional attention weights (*attention*) that sum up to 1 and indicate the relevance of information from different encoder time steps for the forecasts. On average, *attention* is highest for the most recent 5 days leading up to the forecast date, and then rather similar from time step -6 to -64 (Fig. 7a). Notably, forecast start dates in the Spring and Autumn tend to have higher *attention* to the most recent days (Fig. 7b). Furthermore, it is evident how the high *attention* to recent time steps for forecasts starting in March and April, propagates to time steps further back in time over the next forecast start dates. This suggests that the values of the encoder features (WT, AT, P, SD, and DI) in Spring remain relevant for forecast start dates up to 64 days later at the end of June. Given that *attention* weights are influenced by the *importance* of the model's encoder features, there is also significant *attention* variability for models trained with different random seeds



(Fig. S9). Typically, the relative *attention* to time steps further back in time increases when WT and DI have higher
*importance*, whereas *attention* to most recent time steps tends to increase when the model relies more on AT and SD.

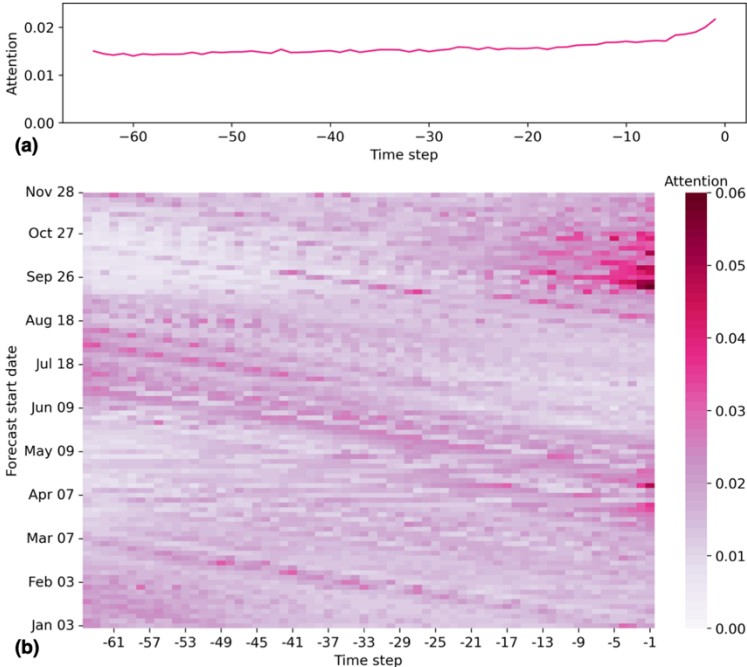

**Figure 7: TFT attention weights to encoder time steps. (a)** *Attention* averaged across random seeds, stations, and forecast start dates as a function of encoder time step. **(b)** *Attention* averaged across random seeds and stations as a function of encoder time step and forecast start date.

In addition to the TFT *importance* weights, here we compare the predictive skill of TFT models with different sets of predictor features (Fig. 8). When omitting the uncertainty of meteorological forecasts, results show a good average CRPS of 0.51 ºC for the simplest model with the following predictors: long-term average and standard deviation of WT (target center and scale) from each station as static features, encoder period WT, and AT. With each new predictor included in the model, the CRPS improves incrementally across almost all lead times, stations and forecasts, reaching an average of 0.39 ºC for the
most complex model with catchment static features, P, SD, and DI as additional predictors. The largest gain in predictive skill is obtained when adding P as a predictor, with the improvement taking place mostly at stations of small and low elevation catchments, as well as being particularly high in Spring. Given that smaller catchments tend to have less streamflow, precipitation events could more easily influence upstream water mixing and consequently station water temperature. Also, in Spring we expect larger water temperature differences between contributing sources such as rainfall
runoff, snowmelt, and lake discharge. In addition to P, the inclusion of DI also clearly improves predictive skill by capturing seasonal characteristics of water and heat fluxes in the catchments. Overall, our findings are consistent with previous studies noting the relevance of precipitation or runoff, as well as global radiation and time of the year for predicting stream water temperature (e.g. Feigl et al., 2021; Zhu and Piotrowski, 2020).



The biases and spread of the ensemble meteorological forecasts limit the gain in predictive skill as more predictors are
included in the TFT models. The average CRPS improves from 0.77 °C for the simplest model to 0.70 °C for the model with
static features, WT (encoder period), AT, P, SD, and DI. Nevertheless, adding SD forecasts is particularly beneficial to
improve model skill for lead times longer than 3 weeks, and at high elevation stations such as 2256 (Rosegbach –
Pontresina), 2269 (Lonza – Blatten), 2612 (Riale di Pincascia – Lavertezzo), and 2617 (Rom – Müstair). Lastly, we note that
including SD and DI is especially beneficial for forecast start dates from mid-April to end of June.




**Figure 8: Predictive skill comparison of TFT models with different features.** The cumulative rank probability score (CRPS) of each
model is shown as a function of **(a,d)** lead time averaged over all stations and forecasts, **(b,e)** station averaged over all lead times and
forecasts, and **(c,f)** forecast start date averaged over all lead times and stations. The CRPS is shown when computed omitting the
uncertainty of meteorological forecasts **(a,b,c)**, and when computed including the uncertainty of meteorological forecasts **(d,e,f)**. The
legend indicates the different models and their average CRPS over all 32 lead times, 54 stations, and 90 forecasts distributed over the year
2022. "AT (no static)" includes long-term average and standard deviation of WT (target center and scale) from each station as static
features, encoder period WT, and AT. "AT" has the same features plus catchment static features (i.e. station coordinates, area, mean
elevation, and glacierized fraction). "ATP" adds P as predictor, "ATPSD" adds SD, and "ATPSDDI" adds DI.



**3.4 Operational forecasts**

Extended range probabilistic forecasts of daily maximum water temperature at the 54 stations in Switzerland are generated operationally twice per week and made available at https://www.drought.ch/de/impakt-vorhersagen-malefix/wassertemperatur-prognosen/. The best performing TFT model with static catchment features, AT, P, SD, DI, and past WT as predictors is used. Figure 9 shows an example forecast for station 2091 (Rhein – Rheinfelden, Messstation) generated on 14 July 2022. Estimates based on each of the 51 ECMWF members provide insight on the expected uncertainty

driven by the meteorological forecasts. The ensuing observations follow closely the best estimate forecast during the first 7 days of lead time and remain within range up to the maximum lead time of 32 days.

It is the aim that stakeholders benefit from our timely forecasts for their decision-making. As a concrete example, an early warning system for fish thermal stress was developed using the stream water temperature forecasts as input

(https://www.drought.ch/de/impakt-vorhersagen-malefix/risiko-von-thermischem-stress-fuer-fische/). High stream water temperatures pose a grave threat to fish populations (Barbarossa et al., 2021), as experienced during the summer of 2018 in Switzerland when tons of fish died in the Rhein. Therefore, timely knowledge on the exceedance potential of dangerous water temperature thresholds as given in Figure 9 is important. The July 14 forecast indicated occurrence probabilities generally greater than 60% for daily maximum stream water temperature to rise above 24 ºC at station 2091 over the coming

weeks, which was then the case from July 14 to August 8.

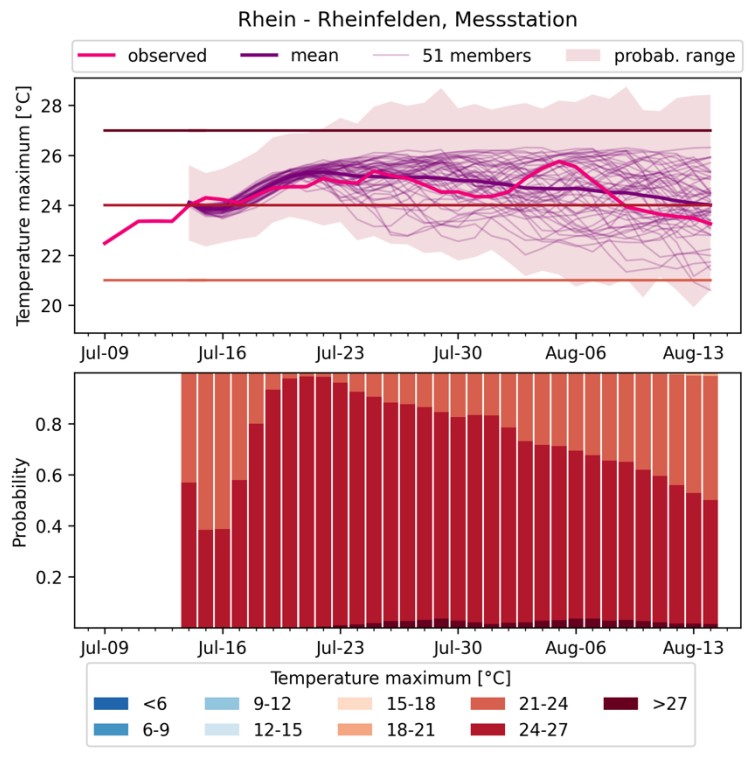





**Figure 9: Operational daily maximum water temperature forecast on 14 July 2022 at station 2091 (Rhein – Rheinfelden, Messstation).** The forecast is generated with the best performing TFT model with static catchment features, AT, P, SD, DI, and past WT as predictors. The temporal evolution of observed water temperature is shown at the top panel together with the forecast best estimate, the 51 estimates based on each of the ECMWF ensemble members of the meteorological forecast, and the probability range. The bottom panel shows the daily forecast probability for different categories of daily maximum water temperature.

## 4 Conclusions

In this study we evaluate state-of-the-art deep learning models for stream water temperature predictions over the next 32 days, given the high value of skillful probabilistic extended range forecasts for managerial decisions. Deep learning models go beyond the iterative predictions of their standard counterparts, by efficiently generating instead direct multi-horizon forecasts across multiple stations at once. The TFT model performs best with an average CRPS of 0.70 °C, degrading from 0.38 °C at 1 day lead time to 0.90 °C at 32 days lead time. We find that 0.31 °C of the average disagreement between observations and predictions stems from the uncertainty in the meteorological forecasts of AT, P, and SD. This is a novel insight on the current limits of extended range water temperature predictability.

The 54 stream water temperature stations from the Swiss Federal Office for the Environment monitoring network comprise catchments varying in size, elevation, steepness, and different degrees of human interventions. These data are exploited by the deep learning algorithms to generalize the relationships between water temperature and its predictors. Our results demonstrate the potential of the TFT model to predict water temperature at stations on which it was not trained, and at ungauged locations. The error at new stations increases 0.11 °C reaching an average CRPS of 0.83 °C, whereas it increases to 1.04 °C when local water temperature is unavailable to the model. This is an important step forward in the quest to expand the availability of stream water temperature estimates across the world. Furthermore, the TFT model may also be used to generate predictions under future climate scenarios assuming no major changes on how water temperature responds to changes in the meteorological drivers.

Our detailed analysis of the importance of different model inputs highlights the roles of WT in the encoder period, AT and DI in the decoder period, and of the station specific long-term average of daily maximum water temperature among the static features. The TFT model with only WT and AT as time-varying inputs performs well with an average CRPS of 0.76 °C, which improves to 0.70 °C when including P, SD, and DI. Overall, the TFT insights on feature importance and attention to encoder period time steps for each forecast provide valuable interpretability that was often missing in machine learning models.

The publicly available operational system for extended range forecasts of daily maximum water temperature completes our contribution by regularly providing information to parties interested on the consequences of extreme conditions. The methodology, insights, and product of this study help address the growing challenges surrounding stream water temperatures



– and consequently water quality – in our warming world. Finally, we underscore the necessity for more and better information on this key environmental variable.

**Code and data availability**

The meteorological data used in this study can be requested from the Swiss Federal Office for Meteorology and Climatology, whereas the water temperature data can be requested from the Swiss Federal Office for the Environment. Scripts used in this study will be available upon final publication.

**Author contributions**

RSP, MZ, and KB conceived the idea and designed the study. RSP performed the analysis and wrote the paper with suggestions from MZ and KB. LB led the implementation of the operational forecasts and provided technical support. KB
contributed to the analysis. All authors read and reviewed the paper.

**Competing interests**

The authors declare that they have no conflict of interest.

**Acknowledgements**

We acknowledge support from the MaLeFix project within the EXTREMES programme of the Swiss Federal Institute for
Forest, Snow and Landscape Research WSL. We acknowledge the Swiss Federal Office for Meteorology and Climatology and the Swiss Federal Office for the Environment for providing the meteorological and water temperature data, respectively.

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
