# Peer review of "Extended range forecasting of stream water temperature with deep learning models"

_EGUsphere, 2024_

## Author Comment (AC2)

**Response to reviewers for egusphere-2024-2591: "Extended range forecasting of stream water temperature with deep learning models"**

*Referee #2:*

*The authors evaluated three deep learning models for predicting daily maximum water temperatures at 54 river stations in Switzerland over the next 32 days. While the paper is generally well-structured, there are several issues that need to be addressed.*

We appreciate the feedback. Please find below our reply to your comments.

1) *Please add an LSTM model as a benchmark.*

We now make it clearer that the RNNED model that we use is a generalization of the LSTM. As mentioned in L138–L140 the RNNED uses LSTMs for the encoder and decoder.

2) *Please provide common evaluation metrics such as correlation coefficient, NSE, and MAE. These metrics help readers compare this study's results with other related work.*

We now include some additional metrics in the supporting information. Please also note that CRPS is equivalent to MAE for ensemble forecasts (L220–L223), and that additional metrics only include information from the predicted q50 quantile. Furthermore, we now put our results in context of other related work according to the performance categories defined by Corona et al., (2024).

Corona, C. R. and Hogue, T. S.: Machine Learning in Stream/River Water Temperature Modelling: a review and metrics for evaluation, *Hydrol. Earth Syst. Sci. Discuss.* [preprint], https://doi.org/10.5194/hess-2024-256, in review, 2024.

3) *Line 25: The website provided by the authors does not seem to be updated in real-time, with the last update on August 30th. Please ensure the data is up-to-date.*

The water temperature forecasts are now up to date and have been operational since 2023. However, end of August 2024 the data provider MeteoSwiss moved the data flow to the new HPC-Alps (see link below). During September 2024 this system was under migration and reconfiguration. We are also currently completing a switch to a new secure data flow replacing an older ftp connection. Due to these challenges of making open science with real time data, the forecasts weren't up to date during review.

https://ethz.ch/en/news-and-events/eth-news/news/2024/09/press-release-new-research-infrastructure-alps-supercomputer-inaugurated.html

4) *Line 31: The use of "e.g." followed only by references seems unconventional. The authors should add specific examples or restructure the sentence.*

We restructured the sentence.

5) *Line 42: Please explain what "right direction" means in this context.*

We rephrased the sentence to make it clearer.

6) *Please explain the significance of 52 in Equation 1.*

We expanded the description of the Date Index (DI). The 52 corresponds to the number of weeks in 1 year.

7) *Lines 114-116: Using single grid cell data for some time series variables is inappropriate. For variables like precipitation and air temperature, the authors should calculate catchment averages rather than point values.*

We now also test our model with catchment averages instead of grid cells co-located with the stations and include the results in the supporting information. We find a similar model performance due to the spatial coherence of temperature, precipitation and sunshine duration for the scales of most of the analyzed catchments. Using catchment averages can benefit the predictive skill for relatively small and steep stations like 2612 (Riale di Pincascia – Lavertezzo) and 2068 (Ticino – Riazzino) with high variability of the meteorological conditions within the catchment, but it can also be counterproductive for large catchments like station 2392 Rhein (Oberwasser) – Rheinau and 2288 Rhein - Neuhausen, Flurlingerbrücke where the meteorological conditions hundreds of kilometers upstream of the station are less relevant than the conditions near the station.

8) *Line 119: The authors use Swiss Federal Office for Meteorology and Climatology data for training, and ECMWF data for the 32-day meteorological forecast as input for water temperature forecasting. These two datasets may be inconsistent in spatial-temporal resolution and data distribution. The authors must validate the effectiveness after switching datasets. Suggestion: Assuming today is August 1, 2024, the authors should use ECMWF data from July 1 to August 1, 2024, for prediction and compare it with actual measurements to verify the model's performance in real forecasting scenarios.*

We rephrased the text to clarify the consistency of the data we use. Both observations and forecasts correspond to gridded products with the same spatial resolution of 2km and daily temporal resolution. The ECMWF meteorological forecasts used as input for the water temperature forecasts are also provided to us by the Swiss Federal Office for Meteorology and Climatology.

The suggestion of the reviewer is precisely how we evaluate the models. For each of the forecast start dates in Table S3, we use the ECMWF data provided by MeteoSwiss as input to generate the predictions, which we then compare against actual water temperature measurements.

9) *The authors use 2022 data for testing, but the model can only predict 32 days at a time. Please explain how a full year of predictions is obtained. Is a sliding window method used? If so, please provide details.*

We expanded the text to clarify this point. We use 90 different forecast start dates (see Table S3) distributed throughout the year 2022. For each of these starting dates, the next 32 days are forecasted and then evaluated against observations.

10) *Line 164: Please explain the meaning of "We use 24 forecast creation times every 15 days".*

We expanded the text to clarify that during the year 2021 we select 23 different dates 15 days apart from each other, i.e. 2021-01-04, 2021-01-19, …, 2021-11-15, and 2024-11-30. On these dates a forecast of water temperature is generated for the next 32 days. The models optimize their parameters during training to minimize the error between these 23 forecasts and observations.

*11) Line 179: The authors tuned parameters for the three deep learning models but used default parameters for ARX, RF, and MLP, which results in an unfair comparison. It is recommended to optimize parameters for all models to ensure a fair comparison.*

The simpler models have very few parameters compared to the deep learning models, and the default parameters of the RF and MLP usually work well. Note that the ARX model has no parameters to be tuned. Nonetheless, we now include in the supporting information an analysis of the performance of the MLP and RF with different parameter values.

*12) In the appendix, table headers should be placed above the tables.*

We modified the captions according to the suggestion.

*13) The meaning of table S3 is not clear. Please explain.*

We expanded the text to clarify that the dates indicated in Table S3 correspond to 90 different points in time during the year 2022 when a forecast of water temperature is generated for the next 32 days. The models are evaluated by comparing these 90 forecasts against observations.

*14) Line 196: Why don't ARX, RF, and MLP use static features? If input data is inconsistent, the comparison loses meaning.*

These simpler models do not use static features because they need to be trained separately for each individual station. Adding constant predictors to the models is unnecessary. On the other hand, the RNNED, NHITS and TFT are trained for all 54 stations at once and therefore also include static features to differentiate across stations.

*15) Figure 2(b) is inappropriate because the x-axis (station ID) has no sequential relationship. It is suggested to use a box plot instead of a line graph.*

We consider that including the lines improves the readability of the plot, compared to the option below without including the lines.

[Figure]

16) *Figure 2(a): For lead times 1, 2, 3...32, does the CRPS refer to metrics calculated from predictions for the entire year 2022? Please explain how a full year of predictions and CRPS values are obtained through lead times 1, 2, 3...32.*

Yes, this is correct. The CRPS values correspond to the average over the 90 forecasts indicated in Table S3 and over the 54 stations. For each of the 90 forecasts there is one CRPS value for each lead time 1, 2, 3, …, 32. The values reported in Fig. 2a are the averaged over these 90 forecasts, and also over the 54 stations.

17) *Figure 2(b): Different forecast start times will result in different CRPS values. Does this mean the results are calculated from each forecast start date to the end of 2022? Please explain the calculation method in detail.*

We expanded the text to clarify how the CRPS is computed for each panel of Fig. 2. In total we compute the CRPS for each of the 32 lead times, 54 stations, and 90 different forecasts (Table S3). Fig. 2a averages the CRPS over all stations and forecasts for each lead time, Fig. 2b averages the CRPS over all lead times and forecasts for each station, and Fig. 2c averages the CRPS over all lead times and stations for each forecast.

18) *Please explain lines 237-239 in detail. What does "omitting the uncertainty stemming from the meteorological forecasts" mean? What specific operation does "when using their observed values instead of their forecasts over the 32 days of the prediction horizon" refer to?*

We expanded the text for clarity. It means that in this case the model uses the actual observed values of AT, P, and SD during the 32 days of the prediction horizon when predicting water temperature, as opposed to using the ECMWF ensemble forecasts of AT, P, and SD as it is done later on for Fig. 3 for example.

19) *Line 311: Please explain the meaning of "90 forecast start dates".*

Here the 90 forecast start dates refer to those of Table S3 that are used to evaluate the predictive skill of the models.

*20) In Section 3.3, RF models can also calculate feature importance. It is suggested that the authors calculate the feature importance of the RF model and compare it with the results of the deep learning models.*

In section 3.3 we focus on the TFT given that it outperforms the other models, and because of its built-in interpretability. For the RF it is also possible to estimate feature importance through other methods such as quantifying the decrease in accuracy when randomly permuting the values of a feature. The larger the accuracy decrease, the higher the importance of the feature. In the case of the RF, the feature importance values are estimated with the data from 2012 to 2021 used for the training. Note also that for the RF there is no encoder and decoder distinction, and that the prediction of the RF is done one day at a time.

Nonetheless, we now include an additional figure in the supplement showing the feature importance estimated from the RF models. Across all stations lagged water temperature is the most important feature, commonly followed by AT and DI, whereas SD and P are the least important.

---

## Author Response (AR1)

**Response to reviewers for egusphere-2024-2591: "Extended range forecasting of stream water temperature with deep learning models"**

*Referee #1:*

*The article investigates three main models on their aptitude of predicting the water temperature at specific locations of rivers in Switzerland. It goes on further to compare these models to a set of three simpler, more traditional ML models (RF, ARX and MLP). These models are evaluated in three distinct settings, namely when they were trained on data from all stations and only on a subset of stations while predicting the water temperature on gauged and ungauged stations. As their predictions, the models provide quantile forecasts and therefore directly a measure of uncertainty which is important in real world applications. In addition of investigating the predictive skills of each of the models, the article also provides an analysis of the feature importance for the best DL model (temporal fusion transformer).*

*All in all this work provides a valuable comparison of multiple model architectures for time series forecasting, probably acting as guidepost for future works.*

We appreciate the overall positive evaluation of our manuscript.

*Main comments:*

*1) The model description (starting at L.123) is a bit crammed and for the three deep learning models one architecture illustration each would go a long way to making later aspects more understandable. The fact that NHITS, RNNED and TFT all use encoder and decoders is not clear, and neither is the fact where exactly they use the encoder normalisation. This, however, becomes important on L. 212 (p. 7) where you describe setup B, i.e., the models trained on 20 stations worth of data less and the swap from encoder normalisers to group normalisers. So the suggestion is to include (maybe simpler versions of) diagrams of the models' architectures, aiding the understanding of the later adaptation to the encoder. Another option would be to detail the encoding process for each model where its architecture is shortly described.*

We appreciate the suggestion. We expanded the text and included an additional figure (Fig. 2) to summarize the characteristics of the forecasting models used in our study. We also clarify why for setup B it is necessary to change from normalizing the models' target variable by the long-term average and standard deviation of each station to normalizing by the average and standard deviation of each station during the 64 days of the encoder period. Using the data from the encoder period to normalize the target variable enables the model to generate predictions at stations not included during the training.

In addition, please note that for each of the 20 ungauged stations in setup C, we normalize WT with data over the encoder period from a similar station within the subset used to train the models (Table S1). During the review process we realized that our code did not do this for the first version of the manuscript, but instead also normalized WT with the average and standard deviation of each station during the 64 days of the encoder period as for setup B. We now corrected this oversight, resulting in an updated version of Fig. 6 in the revised manuscript.

*2) The description of the date index (DI) L.111 leaves the question of why it includes a shift by one month, s.t., DI=0 is approx. at the end of January instead of at the beginning? A short explanation with a reference would be nice here.*

We now clarify this in the text. The index was constructed to be symmetrical: highest at the end of July, and lowest at the end of January. This is done such that June and August, May and September, …, and February and December have the same values. We consider that this is a better representation of climate seasonality in Switzerland than an index without a 1-month shift, for which e.g. July and May, as well as August and April would have the same values.

*3) Lastly, section 2.1 describes the data used for training, with it also mentioning on L.97 that "catchment characteristics" are used. However, a list of which characteristics are considered is only given on L.186, in section 2.3. The suggestion is to also explicitly mention the four static characteristics near L.97.*

We now also mention the catchment characteristics where suggested.

*Referee #2:*

*The authors evaluated three deep learning models for predicting daily maximum water temperatures at 54 river stations in Switzerland over the next 32 days. While the paper is generally well-structured, there are several issues that need to be addressed.*

We appreciate the feedback. Please find below our reply to your comments.

*1) Please add an LSTM model as a benchmark.*

We now make it clearer that the RNNED model that we use is a generalization of the LSTM. As mentioned in L147 the RNNED uses LSTMs for the encoder and decoder.

*2) Please provide common evaluation metrics such as correlation coefficient, NSE, and MAE. These metrics help readers compare this study's results with other related work.*

We now include MAE and RMSE results in the supporting information. Please also note that CRPS is equivalent to MAE for ensemble forecasts (L236–L237), and that MAE and RMSE only include information from the predicted q50 quantile. Furthermore, we now put our results in context of other related work according to the performance categories defined by Corona et al., (2024).

Corona, C. R. and Hogue, T. S.: Machine Learning in Stream/River Water Temperature Modelling: a review and metrics for evaluation, *Hydrol. Earth Syst. Sci. Discuss.* [preprint], https://doi.org/10.5194/hess-2024-256, in review, 2024.

*3) Line 25: The website provided by the authors does not seem to be updated in real-time, with the last update on August 30th. Please ensure the data is up-to-date.*

The water temperature forecasts are now up to date and have been operational since 2023. However, end of August 2024 the data provider MeteoSwiss moved the data flow to the new HPC-Alps (see link below). During September 2024 this system was under migration and

reconfiguration. We are also currently completing a switch to a new secure data flow replacing an older ftp connection. Due to these challenges of making open science with real time data, the forecasts weren't up to date during review.

https://ethz.ch/en/news-and-events/eth-news/news/2024/09/press-release-new-research-infrastructure-alps-supercomputer-inaugurated.html

4) *Line 31: The use of "e.g." followed only by references seems unconventional. The authors should add specific examples or restructure the sentence.*

We deleted "e.g." and just kept the references.

5) *Line 42: Please explain what "right direction" means in this context.*

We rephrased the sentence to make it clearer.

6) *Please explain the significance of 52 in Equation 1.*

We expanded the description of the Date Index (DI). The 52 corresponds to the number of weeks in 1 year.

7) *Lines 114-116: Using single grid cell data for some time series variables is inappropriate. For variables like precipitation and air temperature, the authors should calculate catchment averages rather than point values.*

We now also test our model with catchment averages instead of grid cells co-located with the stations and include the results in the supporting information. In new Fig. S9 we find a similar model performance for the TFT due to the spatial coherence of temperature, precipitation and sunshine duration given the characteristics of most of the analyzed catchments. Using catchment averages improves the predictive skill from relatively small and steep catchments with high spatial variability of the meteorological conditions such as 2612 (Riale di Pincascia – Lavertezzo) and 2068 (Ticino – Riazzino), but it is counterproductive at large catchments such as 2392 (Rhein (Oberwasser) – Rheinau) and 2288 (Rhein – Neuhausen, Flurlingerbrücke) where the meteorological conditions tenths of kilometers upstream of the station are less relevant than the conditions near the station.

8) *Line 119: The authors use Swiss Federal Office for Meteorology and Climatology data for training, and ECMWF data for the 32-day meteorological forecast as input for water temperature forecasting. These two datasets may be inconsistent in spatial-temporal resolution and data distribution. The authors must validate the effectiveness after switching datasets. Suggestion: Assuming today is August 1, 2024, the authors should use ECMWF data from July 1 to August 1, 2024, for prediction and compare it with actual measurements to verify the model's performance in real forecasting scenarios.*

We rephrased the text to clarify the consistency of the data we use. Both observations and forecasts correspond to gridded products with the same spatial resolution of 2km and daily temporal resolution. The ECMWF meteorological forecasts used as input for the water temperature forecasts are also provided to us by the Swiss Federal Office for Meteorology and Climatology.

The suggestion of the reviewer is precisely how we evaluate the models. For each of the forecast start dates in Table S4, we use the ECMWF data provided by MeteoSwiss as input to generate the predictions, which we then compare against actual water temperature measurements.

9) *The authors use 2022 data for testing, but the model can only predict 32 days at a time. Please explain how a full year of predictions is obtained. Is a sliding window method used? If so, please provide details.*

We expanded the text to clarify this point. We use 90 different forecast start dates (see Table S4) distributed throughout the year 2022. For each of these starting dates, the next 32 days are forecasted and then evaluated against observations.

10) *Line 164: Please explain the meaning of "We use 24 forecast creation times every 15 days".*

We expanded the text to clarify that during the year 2021 we select 23 different dates 15 days apart from each other, i.e. 2021-01-04, 2021-01-19, …, 2021-11-15, and 2021-11-30. On these dates a forecast of water temperature is generated for the next 32 days. The models optimize their parameters during training to minimize the error between these 23 forecasts and observations.

11) *Line 179: The authors tuned parameters for the three deep learning models but used default parameters for ARX, RF, and MLP, which results in an unfair comparison. It is recommended to optimize parameters for all models to ensure a fair comparison.*

The simpler models have very few parameters compared to the deep learning models, and the default hyperparameters of the RF and MLP usually work well. Note that the ARX model has no hyperparameters to be tuned. Nonetheless, we now include in Table S3 an analysis of the performance of the MLP and RF with different hyperparameter values.

12) *In the appendix, table headers should be placed above the tables.*

We modified the captions according to the suggestion.

13) *The meaning of table S3 is not clear. Please explain.*

We expanded the text to clarify that the dates indicated in now Table S4 correspond to 90 different points in time during the year 2022 when a forecast of water temperature is generated for the next 32 days. The models are evaluated by comparing these 90 forecasts against observations.

14) *Line 196: Why don't ARX, RF, and MLP use static features? If input data is inconsistent, the comparison loses meaning.*

These simpler models do not use static features because they need to be trained separately for each individual station. Adding constant predictors to the models is therefore unnecessary. On the other hand, the RNNED, NHITS and TFT are trained for all 54 stations at once and therefore also include static features to differentiate across stations as mentioned in L191, L206, and now also indicated in new Fig. 2.

*15) Figure 2(b) is inappropriate because the x-axis (station ID) has no sequential relationship. It is suggested to use a box plot instead of a line graph.*

We consider that including the lines improves the readability of the plot, compared to the option below without including the lines.

[Figure]

*16) Figure 2(a): For lead times 1, 2, 3...32, does the CRPS refer to metrics calculated from predictions for the entire year 2022? Please explain how a full year of predictions and CRPS values are obtained through lead times 1, 2, 3...32.*

Yes, this is correct. The CRPS values correspond to the average over the 90 forecasts indicated in now Table S4 and over the 54 stations. For each of the 90 forecasts there is one CRPS value for each lead time 1, 2, 3, …, 32. The values reported in now Fig. 3a are the average over these 90 forecasts, and also over the 54 stations.

*17) Figure 2(b): Different forecast start times will result in different CRPS values. Does this mean the results are calculated from each forecast start date to the end of 2022? Please explain the calculation method in detail.*

We expanded the text to clarify how the CRPS is computed for each panel of now Fig. 3. In total we compute the CRPS for each of the 32 lead times, 54 stations, and 90 different forecasts (Table S4). Fig. 3a averages the CRPS over all stations and forecasts for each lead time, Fig. 3b averages the CRPS over all lead times and forecasts for each station, and Fig. 3c averages the CRPS over all lead times and stations for each forecast.

*18) Please explain lines 237-239 in detail. What does "omitting the uncertainty stemming from the meteorological forecasts" mean? What specific operation does "when using their observed values instead of their forecasts over the 32 days of the prediction horizon" refer to?*

We expanded the text for clarity. It means that in this case the model uses the actual observed values of AT, P, and SD during the 32 days of the prediction horizon when predicting water temperature, as opposed to using the ECMWF ensemble forecasts of AT, P, and SD as it is done afterwards for Fig. 4 for example.

*19) Line 311: Please explain the meaning of "90 forecast start dates".*

We rephrased the sentence. Here we refer to the 90 forecasts distributed over the year 2022 that we use to evaluate the predictive skill of the models. The start dates of each of these forecasts is given in Table S4.

*20) In Section 3.3, RF models can also calculate feature importance. It is suggested that the authors calculate the feature importance of the RF model and compare it with the results of the deep learning models.*

In section 3.3 we focus on the TFT given that it outperforms the other models, and because of its built-in interpretability. For the RF it is also possible to estimate feature importance through other methods such as quantifying the decrease in accuracy when randomly permuting the values of a feature. The larger the accuracy decrease, the higher the importance of the feature. In the case of the RF, the feature importance values are estimated with the data from 2012 to 2021 used for the training. Note also that for the RF there is no encoder and decoder distinction, and that the prediction of the RF is done one day at a time.

Nonetheless, we now include Fig. S13 showing the feature importance estimated from the RF models. Across all stations lagged water temperature is the most important feature, commonly followed by AT and DI, whereas SD and P are the least important.

---

## Author Response (AR2)

**Response to reviewers for egusphere-2024-2591: "Extended range forecasting of stream water temperature with deep learning models"**

*Referee #1:*

*Regarding Question 2: "Please provide common evaluation metrics such as correlation coefficient, NSE, and MAE," you only added MAE and stated that MAE is equivalent to CRPS. However, this does not explain why you disregarded the other metrics (correlation coefficient and NSE). If possible, please also include these metrics.*

We do not consider it useful to additionally provide estimates of NSE/R2 in our study. These metrics are not designed to evaluate probabilistic forecasts nor to be assessed when dealing with an extended range forecast lead time. It is not clear to us what NSE/R2 values should we even compute. As opposed to the CRPS (and also MAE and MSE) where the error metric is defined only by each pair of observations and predictions ($o_i$ and $p_i$), NSE/R2 additionally require the average of the observations ($\bar{o}$). If we were to compute NSE/R2 for each forecast and station separately, using the 32 points from the 32 days lead time to compute $\bar{o}$, it would be important to note that these results are not comparable to those of other studies using prediction models for one time-step ahead. On the other hand, we could extract only $o_i$ and $p_i$ from the first day lead time for all 90 forecasts and compute $\bar{o}$ from these 90 $o_i$ values. In this case however, we would already expect high NSE/R2 values as we would be capturing the seasonal behavior of the water temperature timeseries.

Overall, we consider that the metrics already provided in the study are sufficient to convey our results. Adding NSE/R2 values might even create confusion for the reader.